# Emotional Intelligence Profiles of University Students with Motor Disabilities: Differential Analysis of Self-Concept Dimensions

**DOI:** 10.3390/ijerph16214073

**Published:** 2019-10-23

**Authors:** Raquel Suriá-Martínez, Juan Manuel Ortigosa Quiles, Antonio Riquelme Marin

**Affiliations:** 1Department of Communication and Social Psychology, University of Alicante, Alicante 03690, Spain; 2Department of Personality, Evaluation and Psychological Treatment, University of Murcia, Murcia 30100, Spain; ortigosa@um.es (J.M.O.Q.); riquelme@um.es (A.R.M.)

**Keywords:** emotional intelligence, self-concept, functional diversity, students, university

## Abstract

*Background*: This study compares the self-concept of students that have motor disabilities with that of students in the normative group. It also considers whether there are EI (emotional intelligence) profiles based on combinations of EI components (attention, clarity, and mood repair). Finally, it analyzes whether there are statistically significant differences in self-concept based on the EI profiles found. *Method*: 102 university students with motor disabilities participated. The age range was 19–33 (*M* = 20.22, *SD* = 4.36). The Escala de Autoconcepto Forma 5 (Self-Concept Scale Form 5, AF5) and the Trait Meta-Mood Scale-24 (TMMS-24) were administered. *Results*: The scores for self-concept were lower in the sample of students with reduced mobility. The cluster analysis also identified three different EI profiles: one group of young people with high general EI scores, one group with high scores for clarity and mood repair, and a last group of students with low EI scores. Finally, the ANOVA showed better self-concept in the group with high scores in the three EI dimensions. The results suggest that better self-concept is associated with a high EI profile. *Conclusions*: It would be interesting to provide programs that consider EI in more depth to strengthen these students’ self-concept.

## 1. Introduction

The presence of students with disabilities in universities has increased in recent years, and studies dedicated to disabilities have proliferated in this sphere. Most authors highlight the fact that in the current university environment, these students have more difficulties than the general population given that, in addition to the problems caused by the disabilities, they also have to overcome architectural [1,2] and curricular barriers [1,3], as well as the attitudinal barriers of their setting [3,4], and their own internal obstacles [5,6].

One variable that is receiving special interest in the context of successful adjustment and adaptation to the university setting is emotional intelligence (EI) [7,8,9].

Emotional intelligence (EI) refers to processes involved in the use, understanding, and management of emotional states of oneself and others to solve problems and regulate behavior. In this sense, there are two large models of EI that explain the basis of this construct. The first model or mixed model considers EI as a set of stable personality traits, socioemotional competencies, motivational aspects and various cognitive abilities [10,11,12]. The second one, called the skill model, would be oriented to people’s abilities to perceive, value, and repair emotions by promoting emotional and intellectual growth [13].

From this last approach, and although there are various instruments for measuring it, one of the most widely used is the Trait Meta-Mood Scale (TMMS), a self-report developed by Salovey, Mayer, Goldman, Turvey, and Palfai [14], which empirically identifies three dimensions of EI: (a) attention to feelings, (b) clarity of feelings, and (c) repair of negative moods. These components are directed at identifying and reflecting on emotions (attention), understanding one’s own emotional states (clarity) and, finally, repairing negative feelings (repair). The literature generally shows that subjects with higher EI (evaluated using the TMMS) report fewer physical symptoms, less social anxiety and depression, better self-esteem, greater interpersonal satisfaction, and more use of active coping strategies for solving problems [15,16,17]. Nonetheless, the involvement of each of the dimensions that comprise EI has a different weighting depending on the construct to which it relates [18,19,20,21].

Focusing on university students, various studies [9,22,23] using self-report measures (TMMS), have found associations between the clarity and repair dimensions with higher levels of empathy, more satisfaction with life, and better quality in social relations. Similarly, studies have observed a relationship in adolescent students between high clarity and repair scores and higher scores for self-esteem, happiness, mental health, and life satisfaction, and lower scores for anxiety and depression [16,20,24]. If we consider attention, the association between this dimension with anxiety and depression is positive [9,15,16,25].

If we consider EI and how it relates to essential constructs for young students with disabilities, an association between subjective wellbeing and EI was found, especially in dimensions of clarity and emotional repair [26]. There is a significant variable among the youth population: self-concept. This construct is defined as individuals’ self-perceptions, which is based on their own experiences with others as well as the attributions they make of their own behavior [27]. Empirical evidence emphasizes the importance of this construct for successfully coping with situations and, in general, for the quality of life of young people with disabilities [7,26,28,29,30].

As the International Classification of Diseases, 10th Revision (ICD-10) [31], acknowledges, it refers to any restrictions or disabilities (as a result of a physical deficiency) in individuals who find it difficult to carry out an activity in the manner or level that would be considered normal in their specific sociocultural situation. This entails the loss or abnormality of either anatomical structure and physiological or physical function leading to motor problems, which inflict numerous repercussions on this group’s lives.

In much the same way, different studies have revealed the effect that motor disability has on the self-concept of adolescents and adults [32,33,34].

Accordingly, the International Classification of Functioning, Disability, and Health [35], underlines the close relationship between disability and self-concept, establishing that disability is the consequence of a person’s interaction with different physical obstacles and the attitudes of her or his surroundings, with negative beliefs or attitudes forming undoubtedly significant barriers to social inclusion [4,26,36,37]. Therefore, in addition to the importance of society’s attitude in determining the integration of people with this type of disability, the individual’s self-perception is also important, given that this is what will shape expectations and objectives and so is the driving force that guides behavior [26,30,38].

In relation to this, Buscaglia [39] suggests that while the self-concept of adolescents and young people with disabilities grows and develops in the same way as that of people who do not have disabilities, its evolution is frequently diminished due to the fact that a disabled person receives negative influences on quite a few occasions while facing social rejection and negative experiences in interpersonal relationships, getting to feel devaluation and frustration. These circumstances result in a greater probability of developing a negative self-concept and, therefore, in the necessity to impact on the young disabled people as a collective that is at risk, firstly, due to the experience of living with disability [40,41,42], and secondly, due to the stage of life in which they are, evolutionary period of the lifecycle characterized by greater differentiation of self-concept since young people face new social and cognitive roles as well as important physical changes that favor the emergence of new self-esteem dimensions. All these changes, along with a greater vulnerability or susceptibility of young people to distort their own image, determine the enormous interest that the study of self-concept arouses in this lifecycle period [43,44,45]. This would explain why, from childhood, young people with disabilities often face rejection by others and negative experiences in their social interactions, as well as the hypothesis that students with motor disability are a group with a greater probability of developing negative self-concept [28,30,37].

Regarding the published literature on self-concept and its possible association with EI in primary school students [46] and in secondary school students [47,48], there are several studies examining the significant differences in EI abilities related to self-concept. Nevertheless, there are very few studies on university students with disabilities [49]. The study of EI in these people can be important when it comes to promoting, in them, adequate control and management of their emotional state. Therefore, identifying if there are different combinations of the components of this construct (attention, understanding, and Reparation of emotions) leading to varying patterns of EI may be relevant for academic and personal success. Similarly, there does not appear to be any evidence of published literature concerning possible differences in self-concept according to different EI profiles or patterns between students with motor disability that include overall self-concept, as well as taking into account the dimensions that comprise it (e.g., social, emotional, academic, etc.).

On the abovementioned basis, this study relies on the importance of developing these youngsters’ self-concept. With the aim of examining this relationship in more depth, this study has the following objectives:(1)To discover whether the mean scores for self-concept of students with disabilities differ from the mean scores for self-concept of the normative group.(2)To identify whether there are combinations of EI components that comprise different profiles in this variable according to the role each dimension (attention, clarity, and repair) has in each EI profile.(3)To analyze whether students’ mean scores in the self-concept dimensions differ depending on their EI profiles.

## 2. Method

### 2.1. Participants

For reasons of accessibility, this study is a cross-sectional case series with a purposive sample of 102 students with motor disabilities (37.30% had a degree of severity of under 33%, 25.50% of participants had between 33% and 64%, and 37.30% had a severity of over 65%). The criteria used to divide the groups according to functionality levels were established in the procedure for recognition, declaration, and qualification of the degree of disability in accordance with the Spanish Royal Legislative Decree 1971/1999, of 23 December. This scale distinguishes three disability degrees depending on actual constraints of the basic personal and instrumental activities in daily life for any kind of disability based on mild, moderate, or severe degree: less than 33% (mild disability that allows the individual to maintain an autonomous and functional independence level); from 33% to 64% (includes permanent impairment that cause moderate disability); 65% or more (includes severe permanent impairment that cause very hard activity limitations). The participants indicated this level according to their Social Security functional diversity certificate.

The students are aged between 19 and 33 and their mean age is 20.22 (*SD* = 4.36). Fifty of them were female (49.02%) and 52 (50.98%) males, distributed across the different training courses (30 first-year students, 36 second-year students, and 34 third-year students).

The normative sample was composed of 6.483 Spanish participants aged between 10 and 62, with 44% of them being men and 56% were women [27].

### 2.2. Instruments

#### 2.2.1. Sociodemographic Questionnaire

In order to discover the sociodemographic details (gender, age, and level of functioning of the students), we designed an ad hoc questionnaire for the study.

#### 2.2.2. Trait Meta-Mood Scale-24 (TMMS-24)

To explore the different EI profiles, we used an adaptation of the short version of the TMMS-48 [14] by Fernández-Berrocal et al. [19]. This version comprises 24 Likert-type items (1 = disagree totally; 5 = agree totally), with a score of between 24 and 120. The items are distributed in three scales: emotional attention (items 1 to 8), emotional clarity (items 9 to 16), and emotional repair (items 17 to 24.) Thus, for example, the emotional attention subscale refers to questions such as “Am I paying much attention to feelings?” or “Do I let my thoughts affect my feelings?” Likewise, items referred to “I almost always know how I feel” or “I can come to understand my feelings” would be included in the emotional clarity subscale. Finally, items such as “Although sometimes I feel sad”, “I usually have an optimistic vision” or “I’m concerned about being myself in good spirits” would be part of the emotional repair subscale. We chose this instrument because its psychometric properties are suitable for a university population [50,51]. Consequently, the authors found relationships of the expected type for the three factors with different variables such as anxiety, depression, and subjective wellbeing. Likewise, the levels of internal consistency were higher than 0.80 (emotional attention, α = 0.84; emotional clarity, α = 0.82; mood repair, α = 0.81). This study’s reliability (α) was 0.83 for attention, 0.81 for emotional clarity, and 0.80 for mood repair.

#### 2.2.3. Self-Concept Form 5 Scale (AF-5)

García and Musitu developed this scale [27], based on Shavelson, Hubner, and Stanton’s multidimensional theoretical model of self-concept [52]. It comprises 30 items that conform to five dimensions of self-concept: F1: academic, that refers to the perception of the student role performance; F2: social, that refers to the performance in social relationships; F3: emotional, referring to emotional state as well as involvement and commitment in quotidian life; F4: family, referring to the integration in the family and F5: physical, which refers to the appearance and physical conditions. These items are arranged on a Likert-type scale from 1 to 5 (1 = totally disagree and 5 = totally agree), and the range of scores is from 30 to 150. We chose this instrument as it was used in earlier studies with a population with functional diversity [53]; likewise, the original version’s scale has a factorial structure which satisfactorily confirms the five theoretical factors that comprise self-concept (51% of the total variance of the scale), as well as adequate internal consistency (Cronbach’s alpha = 0.84). In relation to the psychometric properties of AF-5 for the present study, the reliability analysis showed adequate consistency (α = 0.77). Similarly, the percentage of variance explained is 61.04%.

### 2.3. Procedure

After gaining permission from the university authorities to carry out the research, we asked the sample of students with motor disabilities to participate. To disseminate the questionnaire, we used an online announcement on the university’s virtual campus to notify students who might want to participate voluntarily and anonymously about the research and suggest they participate. The questionnaire was available on the internet and hosted on the university virtual campus for three months. The heading of the questionnaire asked them to take part and explained the objectives of the study as well as guaranteeing their confidentiality and anonymity. The survey took about 20 min to complete. We obtained informed consent from the students who took part in the research before they completed the questionnaire.

We also used snowball sampling through the Student Support Centre, as the authors of the study have links with many students with reduced mobility, and asked them to cooperate by telling other students with reduced mobility at the university about it.

### 2.4. Statistical Analysis

To compare whether there are differences between the self-concept of the sample of students with disabilities and the normative group, we used Student’s *t* test for two independent groups. To do so, we used the mean scores from the scale for the normative group to correlate the scores from the AF5 self-concept scale.

The average scores and the standard deviation of the sample with the normative population were those stipulated by the original study of questionnaire AF-5 [27].

To identify the EI profiles, we used the Two-Step Cluster Analysis tool because of its exploratory character and because it was designed to automatically determine the optimum number of groups or clusters in a set of information.

The EI profiles were defined based on the combinations of the three components evaluated by the TMMS-24 scale (attention, clarity, mood repair).

To determine possible statistically significant differences in self-concept depending on the EI profiles or patterns, we performed analysis of variance in the clusters obtained. We then used the Scheffé method to determine the groups between which there are differences. Finally, we used eta-squared and the standardized mean difference or *d* [54] to analyze the effect size of these differences. We used the SPSS 20.0 statistics program (SPSS Inc., Chicago, IL, USA) to analyze the data.

### 2.5. Ethical Statement

This research is based on the ethical principles and recommendations of the “Declaration of Helsinki” (2000), thus attending to the general principle that the concern for the welfare of the participants has prevailed over scientific interests. Likewise, in accordance with Royal Decree 1720/2007, of December 21, which approves the Regulations for the development of Organic Law 15/1999, of December 13, on the protection of personal data.

## 3. Results

### 3.1. Comparison of Mean Self-Concept Scores for the Study Sample and the Normative Group

When comparing the mean scores on the general self-concept scale and for the dimensions of the study groups with the mean scores of the normative group (Table 1), Student’s *t* test indicated statistically significant differences on the overall scale (*t* = 9.47, *p* < 0.001, (η^2^ = 0.41, *d* = 0.81)). The analyses also showed statistically significant differences in the other dimensions that comprise the general self-concept scale, apart from in the family self-concept dimension (Factor 3) where the mean scores for the study sample are higher.

### 3.2. Identifying EI Profiles

After establishing the maximum similarity within each EI profile and the greatest differences between them, the cluster method established three groups based on the weight of each dimension, namely, attention, clarity, and repair. In this way, we divide the data as a whole into three groups. The first group contains 28 (27.45%) students with low skills in the three EI dimensions (cluster 1); the second group comprises 36 students (35.29%), where clarity and repair skills are predominant (cluster 2); and in the third group, (cluster 3) made up of 38 students (37.25%), high EI skills are predominant (Figure 1).

### 3.3. Differences between Groups in Self-Concept Dimensions

The scores obtained on the overall scale (Table 2) suggested statistically significant differences for the three profiles (*F*_(2, 101)_ = 7.32, *p* < 0.05, η^2^ = 0.32), and we observed that Group 3 (HGEI) had higher means than Group 1(LGEI) (*d* = 0.90), while at the same time, Group 2′s mean (HR–HC–LA) was higher than Group 1 (LGEI) (*d* = 0.71).

When exploring the comparisons in the different self-concept factors, we observed similar results in all of them apart from the physical self-concept factor. Hence, we observed that Group 3, the group with high overall EI scores (HGEI), as well as Group 2 (HR–HC–LA), had significantly higher mean scores than Group 1 (LGEI) for the academic self-concept factor (*F*_(2, 101)_ = 4.24, *p* < 0.05, η^2^ = 0.3), with a moderate effect size found between Groups 3 and 2 when compared with Group 1 (*d* = 0.64–0.56).

Regarding self-concept relating to social interaction, we observed statistically significant differences between the groups (*F*_(2, 101)_ = 3.91, *p* < 0.05, η^2^ = 0.31), and we noted that the students in Groups 2 (HR–HC–LA) and 3 (HGEI) had higher scores than Group 1 (LGEI) (*d* = 0.84 and 0.61).

We also found similar results when examining the mean rates in factors 3 and 4. In this case, in the factor relating to family self-concept, the analyses showed that Group 3 (HGEI) had higher scores than Group 1 (LGEI) (*F*_(2, 101)_ = 6.50, *p* < 0.05, η^2^ = 0.36), (*d* = 0.92), while Group 2 (HR–HC–LA) had higher mean rates than Group 1 (LGEI) (*d* = 0,71). Regarding emotional self-concept, Group 3 (HGEI) stood out with higher mean scores than Group 1 (LGEI) (*F*_(2, 101)_ = 9.58, *p* < 0.001, η^2^ = 0.40, *d* = 0.56), and Group 2 also had higher mean scores than Group 1 (LGEI) (*d* = 0.49).

## 4. Discussion

This work has attempted to explore the relationship between the skills that comprise EI and self-concept in a sample of university students with reduced mobility.

Regarding the first objective: to establish whether there is a difference between the self-concept scores of students with reduced mobility and those of the normative group, the results in general show lower mean scores for students with reduced mobility in both general self-concept and the other factors with the exception of the family self-concept factor. This is in line with previous research [32,33,34]. This fact could be explained because the people with disability are more self-critical, they may be overprotected through their childhood, fall into poorer quality intimate relationships, or get lower-level jobs [33]. Hence, the different types of physical damage (mobility and communication problems, disturbance and loss of sphincter control, sensitivity, sexual response, etc.) can lead to important social and psychological effects that are caused by this disability [40,41,42]. Specifically, this is not reflected in the family self-concept. It is likely that support and strong protection from the family is especially important in these students’ lives, and that their parents, relatives, and sometimes even teachers themselves have strongly protective attitudes towards them, and so these young people perceive and feel this protection [30,36].

Regarding the second objective: to identify whether there are combinations of the EI components that form different profiles for this construct, the cluster analysis identified three different profiles depending on the weight of each skill (attention, clarity, mood repair) in each of the groups. Consequently, we found one group with low scores in the three components of this ability (Group 1: LGEI), another with group low scores for attention and high scores for clarity and mood repair (Group 2: HR–HC–LA), and a final group with high scores in all three EI skills (Group 3: HGEI). These profiles are on the same lines as previous studies. Therefore, different items of research have discussed a positive association between personal, social, and academic variables and high general EI, which possibly relates to the profile defined in Group 3 of our study [8,15,17]. For their part, other works show differing associations between variables linked to adequate interpersonal adjustment and those relating to the EI skills in charge of understanding one’s own emotions, other people’s emotions, and mood repair (e.g., social relations) [9,18,55], which would support the profile where high clarity and repair scores and low attention scores stand out (Group 2). Finally, low scores in the three EI skills are associated with deficiencies in psychological adjustment, interpersonal relationships, and academic success [19,22,55,56], supporting the low overall EI profile (Group 1).

From the evaluation of these results, it is obvious that the clarity and emotional repair skills are essential for a good fit, whereas emotional attention is considered irrelevant and even negative for a successful fit whether the clarity and emotional repair skills are scarce.

Regarding the *third objective*, it was observed that the mean rates for the profiles obtained differ in self-concept depending on the weight of each EI dimension. Accordingly, the data show that students with reduced mobility who have high EI levels have a more positive perception of themselves than students with low EI scores in most self-concept dimensions.

Such results show little or no difference between Group 2 and Group 3. In line with other studies, it seems that high score in emotional attention would be irrelevant or could even have a negative impact in absence of clarity and regulation dimensions. This might be due to the fact that an excessive attention paid to emotions could lead to discomfort and concern in young people [15,18,26].

Hence, when considering the mean scores of the groups, we find that to achieve adequate academic self-concept (Factor 1), it is vital to have high scores in the three EI dimensions (Group 3). These results support those obtained by other authors [8,15] who have found a positive relationship between academic performance and the three EI skills in students in secondary education.

Accordingly, academic work and intellectual development entail the ability to use and regulate emotions to facilitate thinking, increase concentration, control impulsive behavior, and perform under stressful conditions [55]. The repair skill, one of the fundamental components of EI, is especially important in this process [8,9,55].

In summary, we note that in the other self-concept factors (except for physical), the groups with high scores in mood repair and emotional clarity stand out (Group 2 and Group 3), while the attention dimension is not relevant. Hence, different authors emphasize that these skills are essential for the self-concept factors relating to emotions, social relationships, and coexistence (social self-concept and family self-concept) [8,9,26,38].

Finally, the data do not show differences between the three EI profiles and the factor relating to physical self-concept (Factor 5). This could be important because of the fashions and stereotypes imposed by society, as these often exclude people with motor disabilities from stereotypes of beauty [1,4,30,38]. This could explain why regardless of their EI profile, lower levels of self-esteem are more common in these students.

Based on these results, this work firstly shows the deterioration of these students’ self-concept. Without any doubt, until this vital cycle stage, the young people are immersed in their relationships with peers, teachers, and family, who with a likely negative perception, may influence these youngsters’ self-concept and personal growth in a negative way. Similarly, the current society determines a kind of fashion that moves the disabled people from beauty patterns [26,41]. This may have adverse effects, such as lower levels of self-esteem in the individuals of this group. All of this stresses the importance of implementing this construct in family, social, and educational contexts since they can favor a successful adjustment in different spheres if they have a negative perception. At the same time, this research highlights the connection between EI and characteristics that shape the personality, such as self-concept, which will favor a successful adjustment to the environment [13,16,55].

Nonetheless, this study does have some limitations. The main one is that each person will experience this situation differently depending on different aspects (temperament, style of coping, and different contexts), which could result in variability in EI profiles and self-concept in samples with this sort of disability. With regards to methodology, the main difficulty is in conceptualizing EI and how it is measured. There are currently many questionnaires that try to analyze EI, which makes it hard to compare the results of different studies. Future research should control for these biases to increase the internal validity of the results. Another limitation is that there is no control group similar to the experimental one, and it has only been possible to conduct the comparison with the normative scores of the self-concept questionnaire. Future research should control for these biases to increase the validity of the results. In reference to the data provided, these show a university population with motor disability, and they cannot be extrapolated to the remainder of students with different disability.

Finally, given that a pattern with high scores in these clarity and emotional repair dimensions is related to higher self-concept among this group, this work suggests that it could be desirable to design training and development programs for emotional skills such as clarity and mood repair, and so go into greater depth in the relationship between self-concept and EI, thereby enhancing the improvement of self-concept of students with motor disabilities to improve their personal, social, and academic adjustment.

## 5. Conclusions

In conclusion, the study reflects differences in the self-concept of university students with reduced mobility based on their EI profile. Thus, students with high EI perceived themselves in a more positive way than those with low EI in different self-concept dimensions: (academic, social, emotional and family). These results imply the need to develop emotional skills programs aimed at this population in order to favor personal, academic and social adjustment.

## Figures and Tables

**Figure 1 ijerph-16-04073-f001:**
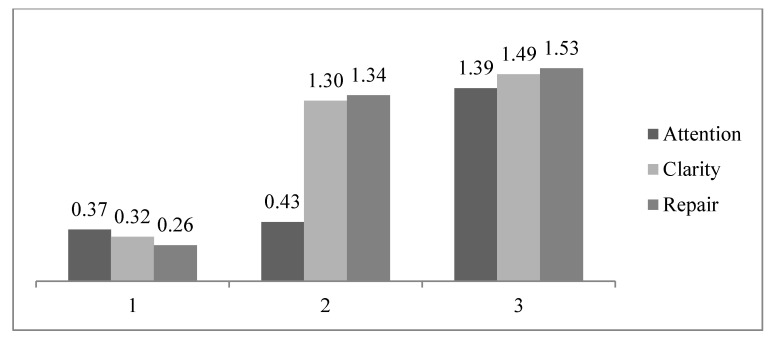
Representation of the three-cluster model: Group 1: LGEI (low general emotional intelligence (EI) scores), Group 2: HR–HC–LA (high repair, high clarity and low attention scores), and Group 3: HGEI (high general EI scores).

**Table 1 ijerph-16-04073-t001:** Mean self-concept scores for the study sample and the normative group.

Self-Concept Factors	Normative Scale		Study Sample		*t*	*p*	η^2^	*d*
	M	SD	M	SD				
F1: Academic	24.02	4.5	23.16	5.92	4.01	0.012	0.18	0.20
F2: Social	23.85	4.46	21.40	5.03	3.48	0.004	0.25	0.51
F3: Family	24.20	3.77	25.57	6.18	6.42	0.042	−0.19	−0.26
F4: Emotional	24.06	4.55	22.65	4.04	4.82	0.000	0.20	0.33
F5: Physical	23.24	5.06	19.24	5.33	2.18	0.000	0.36	0.77
Total scale	120.05	19.25	106.42	14.14	9.47	0.000	0.41	0.81

**Table 2 ijerph-16-04073-t002:** Differences between groups in self-concept dimensions.

Self-Concept Factors	Group 1 Low EI (LGEI)		Group 2 High Repair High Clarity Low Attention (HR–HC–LA)		Group 3 High EI (HGEI)		Total		*F* _(3, 101)_	*p*	η^2^
	M	SD	M	SD	M	SD	M	SD			
Academic	21.21	5.70	23.46	5.99	24.02	4.5	23.16	5.92	4.24	0.010	0.34
Social	19.40	4.57	22.38	5.29	22.85	4.46	21.40	5.03	3.91	0.027	0.31
Family	24.08	4.58	25.78	6.15	25.63	3.77	25.57	6.18	6.50	0.020	0.36
Emotional	21.34	6.13	23.24	7.10	24.06	4.55	22.65	7.04	9.58	0.000	0.40
Physical	19.53	5.74	19.15	4.84	19.24	5.06	19.24	5.33	2.81	0.064	0.11
Total	106.49	23.04	122.15	23.25	125.65	19.25	116.42	24.14	7.32	0.001	0.32

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
