# Peer review of "Emotional Intelligence Profiles of University Students with Motor Disabilities: Differential Analysis of Self-Concept Dimensions"

_ijerph, 2019, doi:10.3390/ijerph16214073_

Round 1
Reviewer 1 Report
This research tested whether EI skills combined to form distinct profiles, and whether classification in each profile was associated with physically disabled students’ self-concepts. Lack of detail in the Methods and Results makes it difficult to evaluate the quality of the work, and a lack of clarity regarding the theoretical rationale and motivations underlying hypotheses make it difficult to evaluate the potential impact of the work.
Introduction and Discussion:
You present two models of emotional intelligence and select the skill model to ground your work. I believe the theoretical motivation for your manuscript would be stronger if you could explain why you believe this model is the more appropriate of the two for this research, and/or if you could discuss how this choice may have influenced your results. In the final two paragraphs of p. 2, you seem to build an argument that people with physical disabilities have an underdeveloped self-concept, and that this leads them to experience negative social interactions and interpersonal rejection. I would strongly consider alternative causal pathways and explanations for the associations among these constructs, both in the introduction and discussion. For example, repeated experiences of implicit or explicit bias in social interactions, perceptions of discrimination in social settings, etc. may hinder development of a positive self-concept. A more nuanced consideration of these possibilities in the discussion may also help to situate your research in the broader context and to identify other potential implications of your results. In the discussion, I would be especially interested in seeing you suggest implications of your research for how other students, faculty, and institutions might better support physically disabled students. I believe the manuscript would be strengthened if you could more fully develop your rationale for why you expect an association between EI and self-concept. Though there may be no existing data on the associations among these constructs in physically disabled youth, you may be able to gain insight from research in other populations. The motivation for the research is not clear. Though you identify that there is a gap in the literature, such that we do not know whether EI profiles are associated with self-concept among students with disabilities, you also do not clearly articulate why answering this question is important. I believe a more nuanced discussion of the limitations and contributions of this research would be beneficial. For example, I would appreciate your expanding on your point about different aspects of individuals’ identities influencing their experiences. Smaller points: P2: “If we consider attention, this association is positive”: It is not clear what is meant by this statement. Presumably, you mean to contrast associations between attention and outcomes with associations between clarity and repair subscales, but the relevant outcomes you are contrasting are unspecified. It would be helpful if you could provide your working definition of self-concept when you introduce the term on p. 2.
Methods and Results:
You describe a comparison between your sample of physically disabled students and a “normative group”, but it is not clear who this comparison group is. Did you collect data from a group of participants without physical disability? Or did you draw values from previous research validating the construct? Please describe in more detail who these data represent and discuss other potential reasons for mean differences in self-concept scores besides the presence vs. absence of physical disability. Several areas of your methods section require further clarification, including: severity of motor disabilities (how is this measured and defined?) “distribution across the different training courses” (Participants section), it is not clear what training courses this statement refers to. Please describe more detail on both the TMMS and the AF-5. It would be helpful to see the number of items along with a sample item from each subscale. Please note whether any other data were collected. Please describe analyses identifying EI profiles in more detail. Some reported effect sizes appear inconsistent with reported means and standard deviations. Please clarify how you calculated partial eta squared and cohen’s d.Overall:
There are several type-o’s and inconsistencies throughout the manuscript. For example, acronyms introduced in section 3.3 are not defined. On page 2, paragraph 3, there is an incomplete sentence: “A significant variable among the youth population: self concept.” There are several places in the results where symbols or values appear to be missing. The manuscript would benefit from careful proofreading, with an eye towards correcting grammatical errors and standardizing use of key terms.
Author Response
We are grateful for the reviewer’s judicious suggestions, which have helped to improve the quality of the manuscript significantly.
Introduction and Discussion:
You present two models of emotional intelligence and select the skill model to ground your work. I believe the theoretical motivation for your manuscript would be stronger if you could explain why you believe this model is the more appropriate of the two for this research, and/or if you could discuss how this choice may have influenced your results.
Since this study intends to examine the relationship between self-concept and Emotional Intelligence (EI) in order to contribute to improve these constructs in this group, we think that understanding EI from a model based on skills will allow the possibility of modifying them, while if we rely on a model of competencies or personality traits of a stable nature, their stability would be more likely over time.
In turn, since one of the aim of the study is to examine the perception of oneself, the use of a self-report tool, such as the TMMS, which examines the perception of the ability of emotional self-regulation, would make more sense along with the Self-concept assessment.
In the discussion section, in addition to the introduction, we have added the point that the use of a self-report can limit results that might have a more than likely bias.
In the final two paragraphs of p. 2, you seem to build an argument that people with physical disabilities have an underdeveloped self-concept, and that this leads them to experience negative social interactions and interpersonal rejection. I would strongly consider alternative causal pathways and explanations for the associations among these constructs, both in the introduction and discussion.
As the reviewer suggests, this information has been expanded in the introduction section. In the discussion section explanations about this association have also been included.
In the discussion, I would be especially interested in seeing you suggest implications of your research for how other students, faculty, and institutions might better support physically disabled students. I believe the manuscript would be strengthened if you could more fully develop your rationale for why you expect an association between EI and self-concept.
According to the reviewer, this information has been expanded in the discussion section.
Though there may be no existing data on the associations among these constructs in physically disabled youth, you may be able to gain insight from research in other populations.
As the reviewer indicates, studies of this relationship in children and adolescents have been included in the introduction section.
The motivation for the research is not clear. Though you identify that there is a gap in the literature, such that we do not know whether EI profiles are associated with self-concept among students with disabilities, you also do not clearly articulate why answering this question is important.
One of the aim of the study is to examine the self-concept, and then check if certain EI profiles are related to the self-concept. In this way, deepening these possible associations could help strengthen the self-concept of these young people through emotional skills.
I believe a more nuanced discussion of the limitations and contributions of this research would be beneficial. For example, I would appreciate your expanding on your point about different aspects of individuals’ identities influencing their experiences.
Information about the influence of family, peers and society on the self-concept of these young people has been included in the Discussion section.
Smaller points: P2: “If we consider attention, this association is positive”: It is not clear what is meant by this statement. Presumably, you mean to contrast associations between attention and outcomes with associations between clarity and repair subscales, but the relevant outcomes you are contrasting are unspecified. It would be helpful if you could provide your working definition of self-concept when you introduce the term on p. 2.
These results show the idea that emotional clarity and repair skills are essential for a good adjustment, however, emotional attention is not relevant, or even it negatively affects this successful adjustment if the clarity and recovery skills are scarce.
The definition offered by the authors of the instrument used in our study (Garcia and Mussitu, 1999) has been included.
Methods and Results:
You describe a comparison between your sample of physically disabled students and a “normative group”, but it is not clear who this comparison group is. Did you collect data from a group of participants without physical disability? Or did you draw values from previous research validating the construct?
The data used in the normative sample of the present study are the same as those used in the original study (García & Musitu, 1999).
Describe in more detail who these data represent and discuss other potential reasons for mean differences in self-concept scores besides the presence vs. absence of physical disability.
In addition to the paragraph of the International Classification of the Functioning, Disability and Health (WHO, 2001) which affects the close relationship between disability and self-concept, a possible explanations of other studies that affect a lower self-concept among this population has been included in the discussion section (Nosek, Hughes, Swedlund, Taylor & Swank, 2003; Tam, 1998; Tam, Chan, Lam & Lam, 2003)
Several areas of your methods section require further clarification, including: severity of motor disabilities (how is this measured and defined?) “Distribution across the different training courses” (Participants section), it is not clear what training courses this statement refers to.
As the reviewer rightly suggests, this information is missing. The participants section includes criteria of severity of disability as well as distribution of participants in the different courses.
Describe more detail on both the TMMS and the AF-5. It would be helpful to see the number of items along with a sample item from each subscale.
According to the reviewer's suggestions, this information has been expanded, including description of dimensions and number of items per subscale.
Please note whether any other data were collected.
Through the Chi-square test for homogeneity of frequency distribution, it was found that there were no statistically significant differences between the groups according to sex, age, severity of disability and academic year.
Describe analyses identifying EI profiles in more detail.
We have described the load of each component in each cluster within Figure 1 and the percentage of sample in each cluster in the text.
Some reported effect sizes appear inconsistent with reported means and standard deviations. Please clarify how you calculated partial eta squared and Cohen’s d.
The effect size was estimated using the coefficient of determination (R2). The Cohen index was obtained by calculating the average difference typified through the statistical program.
There are several type-o’s and inconsistencies throughout the manuscript: acronyms introduced in section 3.3 are not defined.
Every dimension acronym has been included.
There are several type-o’s and inconsistencies throughout the manuscript: On page 2, paragraph 3, there is an incomplete sentence: “A significant variable among the youth population: self-concept.”
The sentence has been completed.
There are several type-o’s and inconsistencies throughout the manuscript: There are several places in the results where symbols or values appear to be missing. The manuscript would benefit from careful proofreading, with an eye towards correcting grammatical errors and standardizing use of key terms.
The paper has been revised following the precious comments. Typos, grammar, semantics, references and abbreviations have been improved for clarity purposes.
Reviewer 2 Report
This study focuses on emotional intelligence and self concept in individuals with motor vehicles. Past research on EI is well-presented and the paucity of literature on EI in this sample population is discussed. The main weaknesses of the present study is that there is no operational definition offered for motor disabilities. How was this defined for the purpose of this study? What was the criteria for study inclusion? How was the degree of severity for motor disabilities determined/measured for this study? The word case series when describing the participants seems inappropriate here as it is not a clinical study. The authors mention that they are comparing their study sample to a normative group - who is the normative group and what is the sample size and profile of the normative group? How is "normative group" defined for this study? Is the normative group profiles from a previous study of EI and self-concept? This is not clear in the methods or analysis. For the three groups with three different EI profiles, were there any demographic similarities (such as a correlation with severity of motor disabilities) between the three groups? In the procedure section, it is unclear if the questionnaires were distributed online or through a paper-based survey - it is not clear if virtual campus refers to an online survey site. There are minor English language editing issues. The authors should strengthen their methodology section and offer clear operational definitions for their study sample and terminologies used, specifically as it relates to a group with motor disabilities and the normative group.
Author Response
We are grateful for the reviewer’s judicious suggestions, which have helped to improve the quality of the manuscript significantly.
This study focuses on emotional intelligence and self-concept in individuals with motor vehicles. Past research on EI is well-presented and the paucity of literature on EI in this sample population is discussed. The main weaknesses of the present study is that there is no operational definition offered for motor disabilities. How was this defined for the purpose of this study?
As the reviewer rightly suggests, this definition is missing. The one provided by ICD-10 refers to ‘any restrictions or disabilities (as a result of a physical deficiency) in individuals who find it difficult to carry out an activity in the manner or level that would be considered normal in their specific socio-cultural situation’. (WHO, 1995).
What was the criteria for study inclusion? How was the degree of severity for motor disabilities determined/measured for this study? The word case series when describing the participants seems inappropriate here as it is not a clinical study.
Following the reviewer’s suggestions, the inclusion criteria has been included in the text: The criteria followed to divide the groups according to the level of functionality was established in the procedure for the recognition, declaration and qualification of the degree of disability of Royal Decree 1971/1999, of December 23 (Spain). Participants indicated this level according to their Social Security functional diversity certificate.
The authors mention that they are comparing their study sample to a normative group - who is the normative group and what is the sample size and profile of the normative group? How is "normative group" defined for this study?
The data used in the normative sample of the present study are the same as those used in the original study (García & Musitu, 1999). The normative sample was formed by 6,483 Spanish participants, of which 44% were men and 56% women, aged between 10 and 62 years-old (Garcia and Musitu, 1999).
For the three groups with three different EI profiles, were there any demographic similarities (such as a correlation with severity of motor disabilities) between the three groups?
This information has been included in the participants section: Through the Chi-square test of homogeneity of frequency distribution, it was found that there were no statistically significant differences between the groups according to sex, age, severity of disability and academic year.
In the procedure section, it is unclear if the questionnaires were distributed online or through a paper-based survey - it is not clear if virtual campus refers to an online survey site.
As the reviewer suggests, it is more clearly described that access to the questionnaires was online, through the university institution.
Reviewer 3 Report
I think the manuscript is relevant to the journal, and its contribution might be significant.
The proposed objective for this study is clearly defined.
Some suggestions are described below.
The normative group of the self-concept questionnaire is used as the control group. It is necessary to describe the characteristics of this control group (age, sex).
The authors point out that there are few studies examining EI in young people with disabilities. It would be interesting to have a synthesis of what has been obtained in these studies.
The result they obtained about the scale of attention not being a relevant dimension is especially important. Within the area of EI we are even observing in the investigations that an excess of attention is negative.
In the discussion, the justification of the results obtained in the first objective are not related to the conducted analyses. The article states "In relation to this latter aspect, the authors of the original scale (Valenzuela-Zambrano & López-Justicia, 2017) state that as age increases, scores in this factor fall. However, this reduction is not apparent in the population with motor disabilities." The analyses have not been carried out according to the age of the participants, therefore age may not be a possible explanation. The second part of the explanation is in accordance with the results obtained.
One of the limitations is that there is no control group similar to the experimental one and it has only been possible to compare it with the normative scores of the self-concept questionnaire. Perhaps this limitation should be pointed out.
As far as the format is concerned, small errors need to be corrected. Line 94 is missing a sign. Line 99, a dot is missing. Lines 196, 203, 207, 211 and 213 are missing a sign. Line 37 and 77: delete "Antonio": References: Line 13, 21. Line 207 “that” appears twice.
References by Raman et al. (2016) and Salguero et al. (2012) only appear in the discussion, not in the introduction. Since they are relevant, it may be necessary to incorporate them in the introduction as well.
In synthesis, I consider that the article is adequate and especially relevant to be published in this journal.
Author Response
We are grateful for the reviewer’s judicious suggestions, which have helped to improve the quality of the manuscript significantly.
The normative group of the self-concept questionnaire is used as the control group. It is necessary to describe the characteristics of this control group (age, sex).
The data used in the normative sample of the present study are the same as those used in the original study (García & Musitu, 1999). The normative sample was formed by 6,483 Spanish participants, of which 44% were men and 56% women, aged between 10 and 62 years-old (Garcia and Musitu, 1999)
The authors point out that there are few studies examining EI in young people with disabilities. It would be interesting to have a synthesis of what has been obtained in these studies.
In this regard, some reference has been added in the text.
In the discussion, the justification of the results obtained in the first objective are not related to the conducted analyses. The article states "In relation to this latter aspect, the authors of the original scale (Valenzuela-Zambrano & López-Justicia, 2017) state that as age increases, scores in this factor fall. However, this reduction is not apparent in the population with motor disabilities." The analyses have not been carried out according to the age of the participants, therefore age may not be a possible explanation. The second part of the explanation is in accordance with the results obtained.
According to the reviewer's observation, that sentence has been removed from the text.
One of the limitations is that there is no control group similar to the experimental one and it has only been possible to compare it with the normative scores of the self-concept questionnaire. Perhaps this limitation should be pointed out.
This suggestion has been addressed in the text.
As far as the format is concerned, small errors need to be corrected. Line 94 is missing a sign. Line 99, a dot is missing. Lines 196, 203, 207, 211 and 213 are missing a sign. Line 37 and 77: delete "Antonio": References: Line 13, 21. Line 207 “that” appears twice.
This suggestion has been addressed in the text.
References by Raman et al. (2016) and Salguero et al. (2012) only appear in the discussion, not in the introduction. Since they are relevant, it may be necessary to incorporate them in the introduction as well.
In response to the reviewer’s request, we consider the opportunity to include these references only in the discussion section in order to support and expand other references already included in the introduction section.
Round 2
Reviewer 1 Report
The authors have addressed many of the points I brought up in the initial review. There are some remaining issues I believe could be addressed to further improve the manuscript.
I believe you misinterpreted my point about describing some elements of your methods section. Regarding disability level of participants, I felt it would be helpful to know the definitions of levels of disability described in the methods. I do not have a clear idea of what severity under 33%, between 33-65%, or over 65% means. I also think you misinterpreted my comment to "Describe in more detail who these data represent and discuss other potential reasons for mean differences in self-concept scores besides the presence vs. absence of physical disability." Are there other differences between your sample and the normative sample that might explain your finding that self concept is lower among those with disability, besides the fact that your sample has physical disability? For example, the normative sample was collected 20 years ago, is it possible self-concept has changed over time? Were respondents in the normative sample similar in age to your sample? Relatedly, I believe you need to be explicit in your statistical analysis section that you compared your sample to means and SDs derived from a previously published study. I don't think it's appropriate to give information about the normative sample in the participants section, this makes it seem as though you collected those data. In reply to “please note whether any other data were collected”, you responded “Through the Chi-square test for homogeneity of frequency distribution, it was found that there were no statistically significant differences between the groups according to sex, age, severity of disability and academic year.” I am not sure what this refers to or what question it is supposed to answer. My original question meant to ask whether the survey you collected included any measures besides TMMS, self-concept, gender, age, and functioning. Causal language is used throughout the manuscript, though the cross-sectional nature of the survey design does not allow for causal interpretation.Author Response
Once again, we welcome the reviewer's suggestions that we hope to address with the following answers and comments.
I believe you misinterpreted my point about describing some elements of your methods section. Regarding disability level of participants, I felt it would be helpful to know the definitions of levels of disability described in the methods. I do not have a clear idea of what severity under 33%, between 33-65%, or over 65% means.
In Spain the criteria to establish the extent or severity of disability are those terms of reference pursuant to the BOE-A-2000-1546 published by the Ministry of Labour and Social Affairs for the purpose of granting economic and social assistance from public administration. This scale distinguishes three disability degrees depending on actual constraints of the basic personal and instrumental activities in daily life for any kind of disability: mild, moderate or severe.
|
Degree of disability |
Description |
|
Less than 33% |
Mild disability that allows the individual to maintain an autonomous and functional Independence level |
|
From 33% to 64% |
Includes permanent impairment that cause moderate disability. There is a big difficulty or impossibility in carrying out some activities. |
|
65% or more |
Includes permanent impairment causing severe limitations for activity. There exists difficulty in some or all of the activities in daily life |
|
75% or more |
Includes severe permanent impairment that cause very hard activity limitations. People affected cannot manage to do everyday activities by themselves. This type involves proven dependence on other people to do the most essential daily activities |
I also think you misinterpreted my comment to "Describe in more detail who these data represent and discuss other potential reasons for mean differences in self-concept scores besides the presence vs. absence of physical disability." Are there other differences between your sample and the normative sample that might explain your finding that self-concept is lower among those with disability, besides the fact that your sample has physical disability? For example, the normative sample was collected 20 years ago, is it possible self-concept has changed over time?
The factors influencing the self-concept are varied from the beauty standards of society until the individual’s own direct experience, such as the professional or social success. With regard to the studies consulted, they provide possible explanation of those differences, as for example, the existence of a greater self-criticism (openness), a greater overprotection throughout childhood, poorer quality of intimate relationships or lower-quality jobs, and all of them are determined by the presence of disability. Following the reviewer’s suggestion those items are commented in discussion on objective 1.
Concerning the possibility that the self-concept might have changed in the course of time, it is possible that some social and cultural changes have led to variations in factors influencing the construction of either self-concept in general or appearance in particular. But as we have already commented, from the psychometrical perspective the dimensionality of the construct still remains without substantial changes throughout the years and across the countries.
At present it has been examined the use of AF-5 scale in the latest psychometrical studies in several Spanish regions (Catalonia, Basque Country, Region of Valencia) and in other previous adaptations (Brasil -Martínez and García, 2008; Martínez, García and Yubero, 2007; Martínez, Musitu, García and Camino, 2003- and Portugal -García, Musitu and Veiga, 2006), in adolescents and university people. For example, Malo, Bataller, Casas, Gras and González (2011), as well as Esnaola, Rodríguez and Goñi (2011), do not find differences in factorial structure, so, they confirm the same multidimensional structure of the construct. Just Esnaola et al. (2011) propose to split the physical dimension into physical attraction and physical condition following Fox and Corbin’s proposal (1989).
Finally, García, Gracia and Zeleznova (2013) adapted this instrument to the English-speaking American population again replicating the structure of 5 factors, the physical self-concept included among them.
Were respondents in the normative sample similar in age to your sample?
The normative sample drawn from AF-5 refers to adolescents and adults aged 12 to 60.
I believe you need to be explicit in your statistical analysis section that you compared your sample to means and SDs derived from a previously published study. I don't think it's appropriate to give information about the normative sample in the participants section, this makes it seem as though you collected those data.
The following information is included in the statistical analysis section:
To compare whether there are differences between the self-concept of the sample of students with disabilities and the normative group, we used Student’s t test for two independent groups. To do so, we used the mean scores from the scale for the normative group to correlate the scores from the AF5 self-concept scale.
According to the reviewer, we transferred the provided information to the section of statistical analysis. Anyway, we consider that placing this information in the section of participants could enhance the reader’s understanding as this descriptive information is thus close to the study sample.
In reply to “please note whether any other data were collected”, you responded “Through the Chi-square test for homogeneity of frequency distribution, it was found that there were no statistically significant differences between the groups according to sex, age, severity of disability and academic year.” I am not sure what this refers to or what question it is supposed to answer. My original question meant to ask whether the survey you collected included any measures besides TMMS, self-concept, gender, age, and functioning.
Unfortunately, we really misinterpreted the reviewer’s suggestion. Our aim was to explore the self-concept and the EI instead of addressing the possible influence of other independent variables though; that is why we only include a short socio-demographic questionnaire to know the age, gender, disability degree and academic course. We include all of them as limitations and future lines of work.
Causal language is used throughout the manuscript, though the cross-sectional nature of the survey design does not allow for causal interpretation.
We consider that this study does not pose causality because neither the objective of the study nor the applied statistic procedures are intended to do that. Nevertheless, after having checked the issue again, in the discussion section we emphasize the results of the study, which even if remarking the relation between clarity and emotional reparation with a greater self-concept, they only suggest the disability’s possible influence on self-concept. The purpose of all of it is giving support to the design of training and development of emotional ability programs to enhance self-concept.
